# Mixed Fungal Biofilms: From Mycobiota to Devices, a New Challenge on Clinical Practice

**DOI:** 10.3390/microorganisms10091721

**Published:** 2022-08-26

**Authors:** Polyana de Souza Costa, Andressa Prado, Natalia Pecin Bagon, Melyssa Negri, Terezinha Inez Estivalet Svidzinski

**Affiliations:** Postgraduate Program in Health Sciences, State University of Maringá (UEM), Colombo Avenue, 5790, Maringá 87020-900, Brazil

**Keywords:** medical device, biotic surfaces, co-colonization, mixed infection, co-habit

## Abstract

Most current protocols for the diagnosis of fungal infections are based on culture-dependent methods that allow the evaluation of fungal morphology and the identification of the etiologic agent of mycosis. Most current protocols for the diagnosis of fungal infections are based on culture-dependent methods that enable the examination of the fungi for further identification of the etiological agent of the mycosis. The isolation of fungi from pure cultures is typically recommended, as when more than one species is identified, the second agent is considered a contaminant. Fungi mostly survive in highly organized communities that provoke changes in phenotypic profile, increase resistance to antifungals and environmental stresses, and facilitate evasion from the immune system. Mixed fungal biofilms (MFB) harbor more than one fungal species, wherein exchange can occur that potentialize the effects of these virulence factors. However, little is known about MFB and their role in infectious processes, particularly in terms of how each species may synergistically contribute to the pathogenesis. Here, we review fungi present in MFB that are commensals of the human body, forming the mycobiota, and how their participation in MFB affects the maintenance of homeostasis. In addition, we discuss how MFB are formed on both biotic and abiotic surfaces, thus being a significant reservoir of microorganisms that have already been associated in infectious processes of high morbidity and mortality.

## 1. Introduction

The protocols adopted in clinical laboratories for the diagnostic confirmation of fungal infections usually recommend the isolation of fungi in pure cultures, preferably recovered from a selective culture media [1,2,3]. Great care is taken in this process to prevent contamination in order to avoid “non-pathogenic” fungi being erroneously viewed as agents of fungal infections, as it was previously thought that only a single species could be an etiological agent in each infectious process. However, this idea has changed due to the significant increase in opportunistic fungal infections caused by fungal species that are widely distributed in nature. An increasing number of studies have also reported the isolation of both bacterial and fungal species, as the agents of infection, from the same biological samples taken from patients in severe condition [4]. Furthermore, the COVID-19 pandemic brought to light concomitant cases of fungal infections, such as COVID-19-associated mucormycosis-aspergillosis (CAMA) [5].

Opportunistic fungi can be found outdoors, in the air, soil and on plants, as well as on indoor surfaces, which is critical in the hospital setting, and colonizing the skin and internal surfaces of the human body (mucosa). Furthermore, asymptomatic cross-transmission can occur between humans via the colonization of medical devices. Additionally, they can attack the immunocompromised population, causing serious and fatal invasive infections. Opportunistic mycoses are fungal diseases that affect people with weakened immune systems such as individuals with HIV or cancer, organ transplant recipients, or users of certain medications. All of these issues lead us to reflect on the etiopathogenesis of the fungal infections, highlighting that most of them are caused by opportunistic fungi such as candidiasis and aspergillosis and, thus, these fungi considered “non-pathogenic” can no longer be ignored in clinical laboratories.

Mycologic diagnoses are mainly made by methods based on culture growth and positivesitive identification through the fungal morphological and biochemical aspects. These techniques have limitations, such as low sensitivity, since the identification of the species is based on growth and fungi are usually fastidious. On the other hand, the use of molecular methods, which are a promising tool, raise the argument that simply detecting fungal DNA does not prove the presence of the agent as the cause of the infection [6,7,8]. Other relevant issues must also be taken into account, for instance, in some fungal infections such as onychomycosis (OM), there are inherent gaps related to the development of “pathogenic” fungi such as the dermatophytes, due to the possibility of inhibiting the growth of possible non-dermatophyte molds (NDM), which are now also considered as potential causative agents of OM [6].

Additionally, even though fungi are ubiquitous microorganisms, few species are truly pathogenic. In reality, most fungal species take advantage of an intrinsic or extrinsic imbalance in the host’s immune system leading to the development of a disease. As these fungal species are present in the human mycobiota, there is often a delay in the diagnosis of infections caused by opportunistic fungi due to the difficulty in determining a threshold between colonization and infection, and the fact that infections typically cause non-specific symptoms [9]. With respect to mixed infections, a consensus for laboratory diagnosis is crucial, as for the correct therapeutic indication each causative agent must be identified. Two variants of the same species can comprise a mixed infection. Thus, it is recommended that multiple biological material samples on independent days be collected for culture, as one species may inhibit the growth of another [10].

Mistakes or delays in diagnosis lead to empirical treatments being used in most cases, which can further hinder the treatment as they bring obstacles such as high resistance and high toxicity [10]. The understanding of the mycobiota present throughout the human body is a new and emerging field that must be thoroughly studied. Thus, through a narrative review, we discuss the clinical importance of fungal–fungal interaction in biofilms naturally formed on biotic and abiotic surfaces.

## 2. Microbiota of Human Anatomical Niches

Numerous microorganisms colonize the human body, including bacteria, archaea, fungi, and viruses, and this complex ecosystem is known as the microbiota. The genetic set expressed by these microorganisms is called the microbiome. Specifically, the fungal population colonizing the host is defined as a mycobiota [11]. The presence of these microorganisms provides the opportunity for physical and chemical interactions between different species, genera, and even different kingdoms. Characterization of fungal communities has evolved, but, much less compared to our understanding of the bacterial microbiome, the “bacteriome”; the knowledge about the integrants of the human mycobiome is less than 1% of that of the microbiome. Nevertheless, it is recognized that fungal ecosystems are fundamental to human health and disease [12]. The mycological study of the fungi present in the human mycobiome is still imprecise and limited, as the identification of the fungi has relied on culture-dependent methodologies that require the ideal conditions and growth times for each species [13]. The best alternative seems to be the molecular methods, which have added several advantages over the other techniques [11,13].

With the richness of fungi in different niches, it is not surprising that they are also present in many human systems. Lifestyle, gender, weight, genetics, vertical transmission, environment, presence of diseases, and immune status are some of the human factors already identified as responsible for such high diversity [11,12,13,14]. It has been found to be increasingly similar to a fingerprint, that is, it is unique in each individual, despite it being possible to identify some similarities in each microecosystem.

Each anatomical niche (inner and outer surfaces) of the host has a variety of fungal species forming its own mycobiota, and neighboring niches appear to have similar patterns. Species involved in the same mycobiome can interact with each other via synergistic and/or antagonistic activities [13,15]. Generally, these fungi are recognized as potential pathogens, as they are commensal while the host is healthy and pathogenic when the individual’s health is compromised [11].

Skin is rich in endogenous fungi, in addition to being the first surface in contact with environmental fungi, constituting a complex mycobiota [16]. Human skin colonization is varied and it is important to understand its role in health–disease relationships. The individual’s age seems to be one of the main factors in this variability [16], while environmental variations and individual hygiene also have an influence [13]. Furthermore, each region of the skin has its own characteristics and its distinct ecosystem.

*Malassezia* spp. has been the most prevalent genus identified on human skin and the species vary according to the anatomical niche. This genus has been extensively studied and has been associated with several dermatological conditions, including pityriasis versicolor, atopic dermatitis, and seborrheic dermatitis [16]. The presence of yeasts belonging to the genus *Candida* on the skin of the hands of healthcare professionals and on surfaces of hospital equipment has an impact on nosocomial infections. Thus, the surveillance in hospital settings is an interesting strategy for the prevention of fungal infections [17]. The *Candida parapsilosis* complex, in particular, is common in intensive care units [18]. These data draw attention to the potential risks of contracting a nosocomial bloodstream infection caused by *Candida* spp., which is the most common invasive fungal infection in hospitalized patients and has high mortality rates [19].

The skin of the feet presents a complex ecosystem, composed mainly of yeasts and dermatophytes, while NDM are present in a smaller proportion. Among the genera already identified in this niche are *Malassezia*, *Candida*, *Saccharomyces*, *Cryptococcus*, *Rhodotorula*, *Epidermophyton*, *Microsporum*, *Trichophyton*, *Epicoccum*, and *Aspergillus* [16,20].

The colonization of the respiratory tract is intense and variable due to the constant inhalation of fungal spores. For a long time, the fungal colonization in the lung and respiratory tract was underappreciated, as fungi were only considered during disease processes. However, studies on the pulmonary mycobiota have allowed us to know which fungi colonize the respiratory tract and has identified a correlation between those present in the oral cavity and those present in the environment [21,22]. Other factors that can contribute to the variability of the species present in this niche include location, weather, and occupational environment [23]. Among the genera found in healthy individuals are *Candida*, *Neosartorya*, *Malassezia*, *Hyphodontia*, *Kluyveromyces*, *Pneumocystis*, *Aspergillus*, *Cladosporium*, and *Penicillium* [14,21,24]. In the disease process, the pulmonary fungal diversity changes, becoming smaller and more stable [21,25]. The lower respiratory tract, particularly the lungs, were once considered sterile due to various misconceptions and misinterpretations of data that gave rise to false dogmas in the field. With the advancement of techniques for identifying the microorganisms in the mycobiota, fungi of the genera *Cladosporium*, *Eurotium*, *Penicillium*, *Aspergillus*, *Candida*, *Neosartorya*, *Malassezia*, *Hyphodontia*, *Kluyveromyces*, *Fusarium*, *Acremonium*, and *Pneumocystis* have since been identified [4,21,26].

The oral mycobiota is a well-known niche, with *Candida* spp. being the most studied [23,27]. It is known that yeasts belonging to this genus are capable of producing a significant biomass, playing a relevant role in the oral health-disease dynamism [27]. They are especially involved in cariogenic processes, where the lower pH favors its proliferation [28]. The role of other possible colonizers, such as *Malassezia*, are not yet clear [27,29,30,31,32]. Other less abundant genera such as *Cryptococcus*, *Aspergillus*, and *Mucor* have been found, especially in severely immunocompromised individuals. These are opportunistic pathogens, which may form reservoirs for systemic infections [33,34].

Little is known about the real role of the fungi colonizing the gastrointestinal tract, as microorganisms are usually only isolated during an infectious process in immunocompromised patients, such as esophageal candidiasis. This infectious process occurs due to an immunological imbalance, which favors the proliferation of fungi from the mouth that subsequently cause adherent plaques along the esophagus [35,36,37]. However, no commensal relationships between fungi and the esophageal mucosa have been found in humans [38,39]. The intestinal mycobiome is also beginning to gain attention [40,41]. However, unlike intestinal bacteria, for which a wide variety of species have been identified, the mycobiota appears to be less diverse. Experimental studies indicate that the intestinal mycobiota of C57BL/6 mice is made up of only 10 fungal species, where *C. tropicalis* and *Saccharomyces cerevisae* were the most prevalent [40,42].

The mycobiota data for the urinary system are varied and in some situations contradictory. In healthy individuals, there is no description of the presence of fungi in the urine, but in patients with interstitial cystitis and bladder pain syndrome there may be *Candida* and *Saccharomyces* [13,43]. The vulvovaginal mycobiota is better known, and it is clear that it can be influenced by factors such as age, hygiene, pregnancy, antibiotic use, and urogenital diseases [12]. Moreover, the presence of specific bacteria also regulates the fungal growth in the vaginal ecosystem [12]. Healthy women are predominantly colonized by various species from the genus *Candida*, which under certain conditions can give rise to symptomatic vulvovaginal candidiasis (VVC), mostly caused by *Candida albicans* [23,44]. Furthermore, other fungi have been described in the of vulvovaginal mycobiome; these include *Cladosporium*, *Pichia*, *Alternaria*, and *Rhodotorula* [12,14].

The high adaptability of fungi allows them to create their own microscopic ecosystems, even on abiotic surfaces such as medical devices, in addition to human ecosystems that are naturally colonized [45], and multispecies can be involved [46]. Such communities are known as biofilms, where fungi organize themselves, protected by a self-produced extracellular matrix (ECM), conferring resistance to the host’s immune system and to environmental stressors. Moreover, these properties confer resistance to antifungal agents [47].

Polymicrobial communities formed on abiotic surfaces have been widely studied, especially regarding their role in host homeostasis, regulating neurological, endocrine, and immune processes [48]. The holobiont, a term used to report the host plus its related microbial communities, have co-evolved and adapted to each other over generations [49]. Although not completely elucidated, it is clear that environmental factors such as the increase in global temperature and urbanization have drastically affected the composition of the microbiota, impacting the human evolutionary process and leading to the development of diseases. In summary, the role of fungi in this holobiont complex is still unclear.

## 3. What Are Mixed Fungal Biofilms and Where Are They Naturally Found?

Relevant interactions between the human body, medical devices, and fungi are depicted in Figure 1. Intercession points of them include the mycobiome, clinical intervention, and fungal colonization, which are centralized on the biofilm formation. Indeed, nowadays, it is known that microorganisms are naturally organized in communities called biofilms. Biofilms are surface-associated mono or mixed communities attached to biotic and/or abiotic surfaces, encased in self-produced ECM, that exhibit phenotypes distinct from those of planktonic (free-living) cells [50]. The diverse surfaces where biofilms are usually found include solid abiotic materials (medical devices and non-medical utensils) and biotic surfaces such as tissues and cells [51]. Biofilm formation is an important virulence factor of fungi and contributes to resistance to host immune responses as well as to antifungal resistance and environmental stresses [52,53]. Several fungal species of filamentous, yeast, NDM, and dimorphic fungi have been described as capable of developing in mixed communities [46,54,55,56]. Synergy among fungi is a cooperative interaction between species that produce an effect not achieved by an individual species alone [55,56]. Resulting infections are generally more severe than those caused by individual microorganisms, leading to increased antimicrobial resistance and prolonging the time required for host recovery [57]. Mixed biofilms are prevalent throughout the human body, both in healthy and diseased conditions. However, the clinical concern regarding the synergies of mixed fungal biofilms (MFB) is that the infection will be more severe and recalcitrant to treatment [55]. 

Biofilm formation by *Candida* spp. has been reported to be responsible for bloodstream infections of hospitalized patients at a rate ranging from 16% to 100%, in addition to rarely existing as a mono-species [55,58]. *C. albicans* and *Paracoccidioides brasiliensis* can coexist on oral mucosa and in the case of lesions caused by *P. brasiliensis*, the presence of *C. albicans* can lead to a more aggressive outcome [54]. 

With regards to superficial mycoses, the recurrence of mixed infections is underestimated, but the most known cases involve an association between dermatophytes and yeasts [59]. However, a study demonstrated that from 2006–2015, 6% (149/2473) of these superficial mycoses were mixed infections. In another case of a skin infection the etiology was reported to be *Trichophyton rubrum*, *Aspergillus* spp., and *Scopulariopsis brevicaulis* [60]. An outbreak among young Buddhist monks demonstrated high rates with 45% (27/60) of mixed colonization by dermatophytes [61]. There are some reports on Kerion celsi, a mixed infectious process, which in one study reported a prevalence of 7.83% (9/115) with the following associations, *M. audouinii* and *T. violaceum*; *T. violaceum* and *S. brevicaulis*; *T. soudanense* and *M. audouinii*; and *T. violaceum*, *T. soudanense* and *T. rubrum* [62]. In another report, *T. mentagrophytes* and *M. canis* were found to be the causative agents. These studies demonstrate that the correct diagnosis, especially in mixed infections, is extremely important for choosing the best therapy [63]. 

OM is a public health problem that affects 10% of the entire world population and shamefully still has an inadequate treatment regimen [64,65]. The etiopathogenesis of OM is closely related to the ability of its agents to form biofilms [66]. For a long time, the isolation of more than one species in samples of nail scrapings was erroneously considered as culture contamination. However, data on the incidence of mixed infections in OM have been a concern [6,8,67]. A molecular study of OM samples from Brazil, Canada, and Israel showed that 39% (84/216) of cases were caused by more than one fungal species [6]. Reinforcing this another study reported that OM in 38.8% (47/121) of patients was due to mixed infections [67]. Unusual associations among fungi causing OM have also been reported, for instance *Trichosporon asahii* and *Rhodotorula mucilaginosa* [68], *Trichophyton mentagrophytes* and *Neoscytalidium dimidiatum*, and *T. rubrum* and *N. dimidiatum* [67]. In OM caused by *T. rubrum* co-infected with an NDM, an alteration typically induced by antifungal therapy in this *T. rubrum* strain was blocked. Therefore, this association probably would contribute to the failure of therapy [6]. Rare species are found in distinctly higher percentages in fungal mixed infections than in general. In summary, mixed nail infections represent a challenge both in clinical diagnosis and in the search for a cure [67].

It is clear that invasive medical devices are susceptible to colonization by microorganisms present in the environment and in the human mycobiota. Knowledge about the formation of fungal biofilms on these devices is still relatively recent [47,55,69]. The biofilm formation can make the device an important source for the spread and development of a serious clinical infection. Unfortunately, there are few studies that demonstrate the natural development of biofilms on the surfaces of medical devices [70], especially regarding the formation of MFB. There are reports that show a facilitation of the development of serious infectious processes from the colonization of devices, such as urinary catheters [71] and central venous catheters (CVC) [72]. Other devices are still in the early stages of their mycobiological characterization [73]. For some devices, such as gastrostomy tubes, there are reports of bacterial colonization, but so far little or nothing is known about fungi [74,75]. There are, however, in vitro studies addressing the interaction between artificially produced MFB and abiotic surfaces, such as dental prostheses [76] and the surface of polyvinyl chloride perfusion tubes [77].

Biofilm formed on endotracheal tubes (ETT) is an early knowledge and frequent event in mechanically ventilated patients. It was shown that once a biofilm formed on the surface of the ETT, it was difficult to eradicate [78]. The colonization of the respiratory tract by *Candida* spp. in critically intubated patients was reported to be high (45%), with *C. albicans* being the most frequently isolated species [70]. These authors evaluated the presence of the yeasts in the ETT, showing that more than 90% of them had fungi strongly adhered, suggesting organization in a biofilm form. Tracheostomy tubes can be made of plastic, silicone, or metal, and bacterial biofilms have been found on most of the tubes evaluated [79] but fungal biofilms naturally formed on these tubes have not been reported yet.

One study reported fungi as the most isolated microorganisms with 67.9% (19/28) from peripherally inserted CVC [80]. *Candida* species were the main agents involved in those MFB [72,80,81]. Catheter-related infections may be precursors of bloodstream infections, which are associated with high mortality rates [23,58]. The problem that MFB represents, mainly when associated with medical devices, is emerging and critical in hospital settings.

Periprosthetic joint infections (PJIs) usually originate from hematogenous dissemination or surgical site infection. Although fungi represent less than one percent of all reported PJIs, the increase in cases described in the last decade draws attention, mainly due to the variability in virulence among fungal species, making treatment unpredictable and challenging. Among the PJIs fungal agents, the *Candida* genus stands out, but they are not limited to these yeasts [82,83]. A systematic review gathered all the cases already published that show members of the genus *Aspergillus* to be etiological agents of PJIs in 11 reported cases [83]. Highlighting a study including 18 PJI cases that occurred between 2000 and 2015, where 10 of them were mixed infections, 9 fungus–bacteria, and 1 fungus–fungus [84]. Despite reports of mixed fungal infections, commonly, authors do not raise a discussion about their importance [83,84].

Millions of people around the world wear contact lenses, and their use is increasing [85]. This way of correcting vision brings benefits to users of all age groups [86]. These devices, however, have also been the target of biofilm formation in direct relation to infectious processes that can lead to blindness, such as mycotic keratitis [87]. Several fungal species have been reported as etiological agents of such infections, where *Fusarium* spp., *Penicillium* spp., and *Candida* spp. are the most prevalent [88,89,90]. Recently, a corneal infection caused by more than one fungal species was reported, however the authors suggested that this unusual finding may have been related to the method of material collection [91]. So, is this finding in fact uncommon or are the diagnostic techniques for these devices still flawed?

Denture stomatitis is a painful inflammation of the oral mucosa associated with the use of dental prostheses. Its etiology is multifactorial, but poor hygiene and the propensity of the material to form *Candida* spp. biofilms are key factors [92,93]. Prostheses are commonly manufactured from an acrylic resin, and may be partial [94] or total [95], depending on the patient’s edentulism level, most of whom are elderly [96]. Salivary proteins, such as mucin, together with this acrylic base, provide an environment conducive to microbial adhesion [97]. New techniques in the fabrication of more comfortable prostheses have been used, but unfortunately these have been shown to be more favorable for adhesion in comparison to the traditional fabrication techniques [93]. In addition, *C. albicans* has been identified as a facilitator in the oral epithelium dysplastic and neoplastic processes [98,99]. Confronted with these issues, there is a concern that dental prostheses serve as reservoirs for opportunistic microorganisms. *C. albicans* is efficient in forming MFB with *C. famata*, *C. tropicalis*, and *C. parapsilosis* [100]. This association has been correlated with the predisposition to recurrent stomatitis. Interestingly, a co-infection of *C. albicans* and *C. glabrata* associated to prosthetic stomatitis in an experimental animal model showed no exacerbation of the candidiasis pathogenesis [101]. 

Intragastric balloons (IGB) are devices created in the 1980s to address the treatment of obesity [102]; they have since been improved and today they are usually made of silicone elastomer [103]. This bariatric technique is based on the anticipation of satiety due to the mechanical occupation that the device performs in the stomach [104]. Despite several studies reporting their safety [105], little is known about the contamination of these devices by fungi that resist the inhospitable environment of the stomach. In 2009, Coskun & Bozkurt reported the presence of a “mass” with a necrotizing appearance on the surface of an explanted balloon from a smoking patient. An anatomopathological examination of that IGB revealed hypha-like structures suggesting *Candida* spp., in addition to bacterial growth developed in the culture from the IGB. Smoking, gastric stasis, and use of proton pump inhibitors have been identified as possible triggers for such colonization. Koptzampassi (2013) observed a colonization by *C. albicans* that changed the morphological aspect of the IGB, reinforcing the suspicion raised previously by Coskun & Bozkurt (2009) [106,107]. New models of this device were launched [108], where the filling of the IGB chamber was performed with air, and these also proved to be susceptible to colonization by *C. albicans*.

Other alterations in IGB, such as hyperinsuflation, hypothetically attributed to the fungi, have also been reported. Hyperinsuflation is a spontaneous increase in the volume of the device while inside the stomach due to fungal fermentation that generates gases, and when trapped in the internal chambers of the IGB [109,110]. Non-*C. albicans Candida* (NCAC) species have also been reported to contaminate IGB, *C. parapsilosis* was isolated of fluid in a case of IGB hyperinsuflation [110]. *C. tropicalis* was also isolated from a hyperinflated IGB in mixed cultures with bacteria [111]. A patient with an IGB that developed hyperinsuflation caught attention for the intense fungal colonization [112]. Three regions of that device, including the adjustment pigtail, were affected by a biofilm formed by *C. glabrata*. Additionally, the authors found the same species of fungi in the patient’s duodenum. A series of cases showed the formation of simple and MFB naturally formed by *Candida* spp. on the external surface of IGB where all biofilms were visible macroscopically [46]. This study evidenced the great ability of these microorganisms to build a robust biofilm, which could justify the morphological changes observed in the colonized devices. Nevertheless, the accumulated knowledge about fungal biofilms formed on IGB is still too incipient. Apparently the orogastric fluid is not associated with hyperinsuflation [113]. Thus, there are many gaps to be filled before a proposal of a new hypothesis regarding the possible clinical implication of these fungi on the implanted-IGB surface can be made. There are questions still without scientific answers, for instance: could the intestinal mycobiota be contaminating the IGB? Does this colonization pose a risk to evolving into an infection? New studies should be made in order to elucidate the origin and consequences of this colonization.

Widely used in clinical routine, urinary catheters and stents are medical devices that range from short-term to long-term use, depending on the medical indication. The presence of a urinary catheter is considered a risk factor for invasive fungal infection, and usage time may be mandatory for colonization [114]. *C. tropicalis* has been the most frequently isolated species from candiduria in hospitalized patients and it is related to the use of indwelling devices [115]. Some *Candida* species colonize medical devices used in the genitourinary tract contributing to VVC development in female users. The use of long-term devices is among the main causes of the increased incidence of candiduria in hospitalized patients [115]. Although not statistically significant, the presence of yeast in samples collected from patients with an intrauterine device (IUD) is greater than women who do not use the device (9.5% vs. 5.6%) [116]. Due to the long-term use of the IUD, the presence of biofilm may be associated with recurrence of vulvovaginal candidiasis (RVVC) [117,118]. An in vitro study showed that of 66 clinical isolates of *C. albicans* isolated from IUD, 45.5% (30/66) demonstrated the capacity to develop a biofilm [119]. Another study revealed that *C. parapsilosis* and *C. albicans* were able to adhere and form biofilms on the copper IUD, further increasing its virulence [117,120]. Scanning electron microscopy images captured from the surface of explanted IUD, corroborate these data, as they showed a dense and organized structure of adhered *Candida* spp. associated to bacteria, in biofilm [118]. Interestingly, women with HPV infection also seem to be more prone to biofilm formation; however, these data were only described for bacteria, totally ignoring the high presence of fungi in the vaginal microbiota [121]. Unfortunately, no search focused on MFB has been found when it comes to IUD colonization.

Although still not reported, artificial nails possibly offer potential for MFB formation and consequently the development of OM. This new esthetic device provides conditions that could favor the installation and interaction of multiple fungal species, for example the spaces between the artificial and natural nails generate a humid environment [122]. The acrylic material of the artificial nails can also allow fungal adhesion and could induce lesions by allergic contact dermatitis [122,123,124]. Furthermore, microtraumas are caused by manicures and pedicures, or provoked by shared soaps, clothes, and shoes [123,125,126,127]. Additionally, the sharing of nail polish can facilitate the transmission of dermatophytes. *T. rubrum* was found to be able to survive on nail polish for 60 days at 25 °C [128].

Thus, we observe that fungi co-habit different anatomical niches and can form MFB on different surfaces; however, there is still scarce research showing these fungal–fungal interactions. In this sense, considering reproducible isolation of two or more fungi from biological samples or abiotic surfaces is of high importance so that we can better understand these interactions for advancing the control and therapy of MFB infections.

## 4. Individual Fungal Attributes Relevant to Interaction and Communication in Mixed Fungal–Fungal Biofilms

Fungal–fungal interactions are poorly investigated. There is some information on how the endophyte fungi relationships affect plant chemical and physical characteristics, highlighting the effects that different fungal species have on some ecological interactions [129]. With regards to human infection, the scarcity is even greater, and usually effects related to mixed infections have been focused on bacterial–bacterial, bacterial–fungal, and bacterial–viral interactions [130]. The omics sciences have been of great importance for the understanding of the development of fungal biofilms, revealing their molecular processes, triggers for development, and how they can be prevented or treated [131,132].

According to Table 1, the most common species reported in mixed fungal cultures is *C. albicans*; coincidentally, it is the most studied due to its ability to form biofilm, both single and mixed. The biofilm involving *C. albicans* is the most well-known and characterized, and transcriptional factors have already been identified as protagonists in its biofilm formation. The bacterial–fungal interaction is accepted and held responsible for relevant human infections, especially affecting immunocompromised patients. The origin of some infections such as in the oral cavity, ear, diabetic wounds, chronic pulmonary infections (such as cystic fibrosis), and the urinary tract have been attributed to polymicrobial biofilms formed on medical devices [133]. Similarly, nowadays it is assumed that biofilm formed on the biological or artificial surfaces are the main cause of candidiasis. *C. albicans* biofilms are well-structured communities consisting of the yeast cells pseudohyphae and hyphae, whose formation dynamic is fully established [134]. It begins with the adhesion of yeast cells to a suitable surface, where several attractive and repulsive forces are involved. Strengthening this initial contact are important cell wall-associated adhesion molecules, known as adhesins. Three main families of adhesins play a major role in mediating adherence during biofilm formation: the agglutinin-like sequence (Als), the hyphal wall protein (Hwp), and the individual protein file family F/hyphally regulated (Iff/Hyr) [57]. This is a crucial moment in the establishment of a MFB, since in this phase the yeast cells are able to quickly aggregate with other fungal species [135]. The next steps are the proliferation, yeast-to-hyphae transition, and ECM formation. These processes are regulated by many transcription factors (TF) including Tec1p and Efg1p, and Hwp1p and Hyr1p for filamentation. Rlm1p is related to biofilm maturation and ECM formation, which provides structural support and protection against antifungals and the host immune system.

The interactions of *C. albicans* with other microorganisms can occur via co-aggregation and co-adhesion. Moreover, in both cases an important process is cell–cell communication, which occurs by quorum-sensing molecules and by the release of extracellular vesicles (EV). In a polymicrobial biofilm, the EV are different from those released in a biofilm of a single species. Another difference is that it happens not only for communication, but also for sharing of important resources in the microbial community [143,144]. This interaction can be influenced by altering the extracellular content due the degradation or production of new components. The expression of quorum sensing-dependent genes can be altered, changing the extracellular polymeric substance composition [133]. At this moment, there is no consensus as to whether the intra-biofilm interaction of different fungi is beneficial for or harmful to one or all of participants, since the few publications available are, at times, contradictory.

Throughout the process there is differential expression of the many genes [145]. There is greater expression of the *Hwp1* gene, responsible for adhesion, invasion, and cell wall integrity of *C. albicans*, when associated to *C. glabrata*, although this was not observed in the monospecies biofilm of *C. albicans*. *Als3* has been shown to facilitate the binding of *C. glabrata* to *C. albicans*, adding greater robustness and thickness to the MFB [145].

***Impact of the fungal interaction on virulence.*** Among the partners of *C. albicans* mentioned in Table 1, *C. glabrata, C. tropicalis*, and *C. krusei* stand out. During the formation of a MFB involving *C. albicans* in association with *C. glabrata* or *C. krusei* on a polystyrene surface it was observed that there was a decrease in the number of *C. albicans* viable cells [138]. Indeed, it has been widely demonstrated that an antagonistic interaction occurs when *C. albicans* is associated with a NCAC [146]. A survival curve of a *Galleria mellonella* moth model following inoculation of *C. albicans* and some NCAC species decreased mortality rates to below 35%, while the inoculation with *C. albicans* alone resulted in 100% mortality after 18 h. Results from a murine oral candidiasis model corroborate these data and theorize that virulence of *C. albicans* is modulated in mixed infections and MFB due to competition for adhesion area, filament inhibition, and harmful substances secreted by the species in association [138]. Authors also have mentioned the scarcity of nutrients due to competition between species [146]. In this way, it is necessary to identify the good and common competitors among themselves, for example *C. krusei* was more efficient in inhibiting the development of *C. albicans* as a biofilm former, reducing its cell viability [146]. On the other hand, there was a competitive interaction between *C. albicans* and *C. glabrata* with no expense to the MFB virulence [135]. In this case, a synergistic action was suggested, as while *C. albicans* provided structural support for the attachment of *C. glabrata* cells [135,145], *C. glabrata* received assistance in remodeling its cell wall, mainly by masking the ꞵ-glucan polysaccharides [135]. It was also observed that there was a higher development of the hyphae of *C. albicans*, while there was a greater number of cells of *C. glabrata* during the production of the MFB [145]. 

The knowledge on the contribution of *C. glabrata* in an MFB is still in its outset; however, studies have already revealed genes that participate in the expression of adhesin proteins. The study of these genes provides information, such as in which environments these are more or less expressed. For example, when in the presence of nicotinic acid, which is found in the urinary tract and may be related to biofilms on urinary catheters, the expression of the *Epa6* gene of *C. glabrata* is promoted [147]. Several virulence-related proteins, such as epithelial adhesins, yapsins, and moonlighting enzymes, have been described as components of the *C. glabrata* biofilm ECM [148]. Transcription factors Tec1 and Ste12 were associated with the high adaptability of *C. glabrata* to the different niches of the human body, conferring the ability to resist low pH, high temperatures, and form biofilms [149]. A higher prevalence of *C. glabrata* in biofilm formed on IGB was demonstrated, both alone and in association, and it is noteworthy that such biofilms were naturally formed in devices inserted in the stomach [112].

The behavior of *C. tropicalis* in an MFB is practically unknown, but this deserves great consideration as it is able to produce highly interactive biofilms, and the dissemination ability of this species after a polymicrobial biofilm with *Staphylococcus epidermidis* is augmented. Yeast cells dispersed from monospecies biofilms produced by *C. tropicalis* on catheter were able to migrate and form a fresh biofilm on HeLa and HUVEC human cell lines [150]. Additionally, such yeast cells have a strong influence on the pathogenesis of candiduria in mice [151].

Likewise, *R. mucilaginosa* and *T. asahii* cells recovered from a MFB were more efficient in causing death of the *Zophobas morio* larvae. Apparently, *R. mucilaginosa* was the main beneficiary of the interaction, ensuring greater virulence potential after cultivation in MFB [56].

***Resistance.*** Usually, fungi organized in biofilms are extremely difficult to eradicate, since the growth of aggregates and ECM production are properties that impact the ability of the host to respond to infection. Besides this, the cells that disperse from biofilms display a phenotype of enhanced pathogenicity [53]. Structural complexity, the presence of ECM, metabolic heterogeneity, persister cells, and the regulation of efflux pump genes are all factors that contribute to the MFB being more resistant. Furthermore, the increase in resistance varies with both the drug and the species involved in the MFB [57,152]. Due to the natural resistance or reduced susceptibility of some fungi against antifungals, such as *C. lusitaniae* and *T. asahii* that are resistant to amphotericin B, *C. glabrata* that has reduced susceptibility to azolics, multi-drug resistant *C. auris*, *Aspergillus fumigatus* which is relatively resistant to itraconazole and caspofungin, make the association of these fungi in MFB a dangerous and deadly microbial community [55,57,152].

The three classes of contemporary antifungal drugs suffer a reduction in their efficacy during the growth of the *A. fumigatus* biofilm. This is due to the self-induction of hypoxic microenvironments, which promotes the maturation of the biofilm and increases the resistance to antifungal agents. Such mechanisms have already been related to many bacterial biofilms, but this relationship between oxygen gradients and antifungal resistance has been recently reported [153].

Antifungal resistance in *C. albicans* is multifactorial, and the expression of efflux pumps is the main cause of resistance. The ATP-binding cassette (ABC) and the superfamily of major facilitators (MF) are the main class of efflux pumps, in this type of pump where transmembrane transport occurs due to ATP hydrolysis. The genes that re-regulate their expression are *Cdr* for ABC and *Mr* for MF. Phylogenetic studies have shown that such genes are overexpressed in *C. auris*, in addition to other species that are present. Several other transcription factors are present in *C. auris*, such as *TAC1* and *MRR1*; these molecular rearrangements may be the justification for such resistance found in this species [154].

Persisentsis cells are a controversial issue related to *C. albicans* biofilms and are also involved in the difficulties in eradicating biofilms. There are defined as a subpopulation of metabolically quiescent biofilm cells, express high levels of alkyl hydroperoxide reductase 1 (Ahp1p), which prolongs survival after amphotericin B exposure [155]. Biofilms on biotic surfaces with mutant *C. albicans* cells have a high incidence of persistent cells, suggesting that persistent cells may be associated with persistent colonization or relapsed infection status in vivo [156]. In fact, some studies suggest that persistent cells are not a common feature of *Candida* spp. biofilms [157,158].

Among the main features within fungal resistance in biofilms, the ECM is involved, which is composed of extracellular polymeric substances, exopolysaccharides, nucleic acids, proteins, lipids, and other biomolecules and plays an important role in resistance and tolerance to immune cells [159,160]. The presence of these components slows down the diffusion of antifungal agents inside the biofilms, and some compounds of the ECM, such as mannans and glucans, can interact with the drugs, inhibiting or reducing their action [159].

An in vitro study showed that the combination of caspofungin and fluconazole on an MFB of *C. albicans* and *C. glabrata* caused changes in fungal cell morphology but was not effectively able to decrease the number of viable cells cultured after exposure of the biofilm to drugs. Correlating these data to the high minimal inhibitory concentration (MIC) demanded by cells after biofilm, it can be said that the formation of a simple biofilm increased the drug concentration to 25 times the level necessary for its efficiency, concluding that the MFB can require much more of the compound [161].

The current overview joined many reports and published information, evidencing the existence of mixed fungal biofilms formed on biotic and abiotic surfaces. This topic was presented as a broad perspective. Among the limitations, we highlight the lack of information about the mechanisms involved in the fungus–fungus interaction organized in mixed biofilms, which should be of great interest considering the complexity and versatility of these microorganisms. Furthermore, the available information is practically restricted to the *Candida* genus, due to absence of studies performed with other fungi.

## 5. Conclusions and Perspectives

There is still little knowledge on the fungal–fungal relationship, as the literature does not provide strong evidence for a mutualistic or beneficial effect among fungi involved in MFBs, and the prevalence of mixed infections is underestimated [135]. The lack of standardization in research on the mycobiome and mycobiota makes comparing studies difficult. In addition, most of the existing reports involve a small sampling population of individuals; thus; further investigations, through multicentric studies, are needed, with larger cohorts and in healthy individuals. However, it must be pointed out that it is still difficult to estimate the “health” of a mycobiome [162]. Therefore, deepening this issue is needed and should occur in the coming years.

The first and most urgent line of investigation that needs to evolve is to better understand the fungal–fungal interactions in MFB. For instance, *Fusarium oxysporum* is an important agent isolated from OM, whose ability to form organized and dense monospecies biofilms has already been demonstrated [163]. It has the potential to form in vitro MFB with *C. albicans* [142] but, it was not found in naturally occurring MFBs. As the studies on fungi are later than those on bacteria, there is a tendency to generalize and adopt the protocols known for bacteria, but this is questionable since fungi are eukaryotic microorganisms, with completely different cellular physiology and structure [164]. Fungi should be evaluated by appropriate protocols that meet the specific characteristics of these versatile microorganisms, and variables such as culture media, times, and biomaterial surfaces must be better evaluated and standardized in order to adapt the protocols to the particular characteristics of fungi and to allow specific comparisons between them.

The search for new antifungal treatment options is a promising field that should sustain research in the coming decades. A candidate for a new antifungal drug must present a high ability for penetration and permeation in the MFB, and it should be able to act on each species individually and on the collective community phenotype acquired after interspecies interaction. Additionally, a challenge will be the suitability of laboratory methods addressed to the correct antifungal evaluation. The currently available methods are flawed, as they have not been tested on fungi organized in biofilm [165]. Determination of the MIC and minimum fungicidal concentration (MFC) on fungi in planktonic form certainly do not provide sufficient information on the susceptibility/resistance profile of a potential candidate antifungal drug. Channeled research efforts are needed in order to replace the current approval protocols by other, more suitable ones, where the antifungal active principles are evaluated on fungal biofilm, especially on MFB. Thus, the minimal biofilm inhibitory concentration (MBIC) and minimal biofilm eradication concentration (MBEC) would be more faithful to reveal an interaction similar to that found in an infectious process [166]. Furthermore, in a clinic, the use of drug combinations as an alternative treatment can be important and has already been suggested for an MFB of *C. albicans* and *F. oxysporum* in the *Galleria mellonella* infection model [142]. The pharmaceutical formulations also need to be improved; for example, topical treatment should be considered in the development of new antifungals for OM treatment [64], while antifungals in adhesive formulations seem suitable for the topical treatments of denture stomatitis [167]. 

Other perspectives for the future are the development or search for new materials with the potential to prevent fungal infections, inspired by successful experiments proposed for the prevention of bacterial infections. Likewise, modifications of the hydrodynamic characteristics of existing ones could be evaluated and stimulated for the production of medical devices used in the clinical routine, with a focus on their anti-MFB properties. A new urinary catheter design has been recently proposed, combining chemical, mechanical, and topographical elements, which reduced the occurrence of bacterial urinary infection [168]. With regards to buccal infections, a promising strategy has been the incorporation of compounds with antibiofilm properties into the composition of traditional resin. This was achieved by adding a highly hydrophilic molecule, methacryloyloxyethyl phosphorylcholine, which has the ability to repel proteins and prevent microbial attachment [169]. In that line of thought, PJIs could be forewarned by a silver-coated titanium treatment, since an in vitro evaluation showed promising results in the prevention of *Candida* adhesion on titanium [170]. These authors concluded this simple procedure can be a great option in oncological musculoskeletal reconstruction surgeries, highly relevant in patients with highest risk of infectious complications due to immunosuppressive treatments. Likewise, silver nanoparticles were added to dental resins, and were effective in reducing *C. albicans* biofilm growth [171]. 

In summary, it is clear that the knowledge about MFBs is just beginning, as much remains to be learned about the interactions among two or more eukaryotic microorganisms (fungi). To decipher this complex and important issue, more specified studies are needed to understand MFBs and their role in the human health/disease process.

## Figures and Tables

**Figure 1 microorganisms-10-01721-f001:**
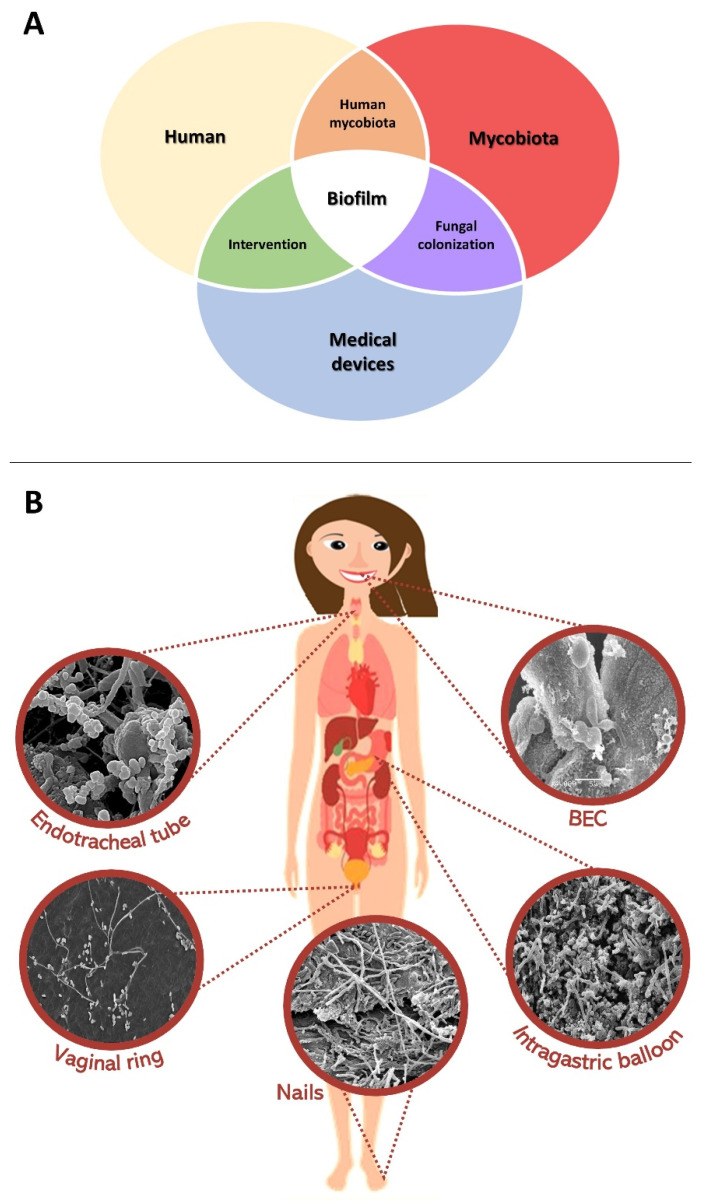
Highlights on mixed fungal biofilms and clinical implications. (**A**) Schematic summarizing the interaction between the mycobiota groups, the human body, and medical devices and their important relationships. The intersection points demonstrate the result of the interaction between each large group for clinical intervention, revealing mycobiomes and fungal colonization, and all situations have a common origin: the formation of biofilm. (**B**) Scanning electron microscopy microphotographs illustrating mixed fungal biofilms formed on biotic and abiotic surfaces correlated with the human body and medical devices such as endotracheal tube, buccal epithelial cells (BEC), intragastric balloons (IGB), nails, and vaginal ring. Mixed fungal biofilms are recognized through previous mycological identification that was performed at all sites; in SEM, the fungal variety is recognized by the presence of several concomitant fungal structures in the same sample, such as blastoconidia and hyphae.

**Table 1 microorganisms-10-01721-t001:** Evidence of mixed fungal biofilm formed in biotics and abiotic surfaces.

Surface	Type of Study	Research Objective	Fungi Involved	MFB Highlights	Ref
Skin	Clinical	Authors discuss the reports of antifungal resistance from around the world, present their experience with treatment-resistant infections, and examine alternative treatment strategies	*Trichophyton rubrum, Aspergillus* spp. and *Scopulariopsis brevicaulis* Dermatophyte and NDM	Therapy resistance probably accentuated due to the presence of mixed-infection-associated drugs, meaning a long treatment time and recurrence after the end of the therapy	[60]
Scalp	Clinical	An analysis of the clinical features and laboratory findings associated with a *Tinea capitis* infection outbreak in young novice Buddhist monks	*Trichophyton mentagrophytes* and *Microsporum canis*	*T. mentagrophytes* and *M. canis* were the predominantly isolated mixed dermatophyte pathogens and an extensive area of infection was significantly associated with mixed-type clinical presentation	[61]
Scalp	Clinical	A clinical case of white scaly alopecia on the scalp of prepubertal children	*Trichophyton violaceum* (violet) and *Trichophyton violaceum* (white)	Before treatment with griseofulvin, *T. violaceum* (white variant) was isolated. After treatment, *T. violaceum* (violet variant) was isolated, indicating the possibility of mixed infection with both variants of *T. violaceum*: white and violet	[72]
Scalp	Clinical	To describe clinical manifestations of *Tinea capitis* in children in southwestern Uganda and identify the main pathogen	*Microsporum audouinii* and *Trichophyton violaceum* *Trichophyton violaceum* and *Scopulariopsis brevicaulis* *Trichophyton soudanense* and *Microsporum audouinii* *Trichophyton violaceum, Trichophyton soudanense* and *Trichophyton rubrum*	Several fungal species known to be pathogenic were found in association, affecting a group of patients with a developing immune system. In addition, in this country, there is a difficulty in the treatment due to there being only a few medication options	[136]
Nails	Clinical	A case of co-habitation of fungus-fungus as causative agents of onychomycosis in a healthy male	*Trichosporon asahii* and *Rhodotorula mucilaginosa*	The case report revealed the presence of dermatophyte and non-dermatophyte in the toenail, highlighting the co-habitation of *T. asahii* and *R. mucilaginosa* in the causation of onychomycosis and to raise the awareness of this infection among dermatologists	[10]
Nails	Clinical	To investigate the clinical manifestations, risk factors, and treatment outcomes of mixed-infection onychomycosis	*Trichophyton mentagrophytes* and *Neoscytalidium dimidiatum* *Trichophyton rubrum* and *Neoscytalidium dimidiatum*	The time of oral treatment for the mixed-infection group was significantly longer than that for the dermatophytes group	[67]
Nails	Clinical	The authors aimed to evaluate the feasibility of introducing microbiological techniques in the diagnosis of nail diseases based only on clinical parameters	*Candida albicans* and *Trichophyton rubrum* *Candida albicans* and *Trichophyton mentagrophytes*	Not surprisingly, *C. albicans* was the most isolated species causing finger nail onychomycosis, but a fact that deserves importance is its association with dermatophytes, mainly due to a diagnostic that demands a lot of experience and care from the laboratory mycologist	[62]
Nails	Clinical	To prove a clinical case of a mixed onychomycosis infection of a toenail	*Chaetomium globosum* and *Trichophyton mentagrophytes*	This association was proved for the first time	[68]
Nasal cavity and paranasal sinuses	Clinical	To report 10 cases of mixed invasive fungal in COVID-19 patients and their outcomes	*Rhizopus arrhizus* and *Aspergillus flavus* *Rhizopus arrhizus and Aspergillus fumigatus*	Mixed fungal infection must be valued, correctly identified and treated, reducing comorbidities for the COVID-19 patient	[137]
Oral	In vivo and in vitro	To evaluate the interaction of MFB in vitro, in vivo with murine models of experimental candidiasis and *Galleria mellonella* larvae	*Candida albicans* and *Candida krusei* *Candida albicans* and *Candida glabrata*	Single infections by *C*. *albicans* were more harmful for animal models than mixed infections with NCAC species, suggesting that *C*. *albicans* establish competitive interactions with *C*. *krusei* and *C*. *glabrata* during biofilm formation	[138]
Endotra- cheal aspirates	Clinical	Reported a respiratory tract colonization of *E. dermatitidis* in a cancer patient suffering from *C. krusei* fungemia and pulmonary disorder	*Exophiala dermatitidis* and *Candida krusei*	The patient’s death was attributed to cancer associated with *C. krusei* fungemia, but probably *E. dermatitidis* also played a role in the morbidity of the case	[139]
Lung	Clinical	This study showed a case of pulmonary co-habitation of two fungal species, *T. mycotoxinivorans* and *C. neoformans*	*Trichosporon mycotoxinivorans* and *Cryptococcus neoformans*	Case of co-infection of the lung with *T. mycotoxinivorans and C. neoformans*. This is the first report of *T. mycotoxinivorans* respiratory infection in Japan	[5]
ETT	Clinical	Aimed to determine the frequency of yeast colonization in the tracheobronchial secretions of critically ill intubated patients and to assess the presence of these yeasts in the infra-cuff region of the ETT	*Candida glabrata* and *Candida tropicalis* *Candida glabrata* and *Candida albicans*	NCAC species were found in co-colonization conditions. More than one species was isolated from both tracheobronchial secretion and ETT in 25% of the patients colonized by *Candida* spp.	[70]
IGB	Clinical	Determine the frequency of biofilms naturally formed on the external surface of IGB, as well as some variables related to IGB types and patients features, species of fungi involved and biofilm evidence	*Candida glabrata* and *Candida albicans* *Candida albicans* and *Candida krusei* *Candida glabrata* and *Candida tropicalis* *Candida glabrata, Candida tropicalis* and *Candida krusei*	Several highly pathogenic fungal species were found, forming mixed biofilms highly adapted to a hostile environment	[46]
Urinary catheter, stent urinary, and urine from these devices	In situ and in vitro	To detect *Candida* spp. using molecular detection by capillary electrophoresis	*Candida albicans* and *Candida parapsilosis* *Candida albicans* and *Candida robusta* *Candida albicans* and *Candida krusei*	The f-ITS2-PCR-CE method was more sensitive and more specific than routine culture both in mono and poly species in the *Candida* colonization	[140]
Intravascular catheter	In situ and in vitro	To search mixed fungal biofilm formed on the intravascular catheter	*Candida albicans* and *Candida glabrata*	*C. albicans* and *C. glabrata* can competitively and symbiotically coexist in a mixed biofilm	[71]
Hip prosthesis	In vivo	To illustrate that *Acremonium* and *Penicillium* species are being increasingly recognized in periprosthetic joint infections	*Acremonium* spp. and *Penicillium* spp.	The authors gave due importance to the growth of commonly neglected fungi and were able to importantly report the positivity of intraoperative mixed fungal cultures	[141]
Polystyrene and *Zophobas morio* larvae	In vitro and in vivo	To evaluate the pathogenesis of a co-infection by two less common yeasts on *Z. morio* larvae	*Rhodotorula mucilaginosa* and *Trichosporon asahii*	Yeasts cells recovered from an in vitro biofilm provoked increased death rates of larvae infected for mixed suspensions	[56]
Polystyrene	In vitro	To investigate the interaction in a dual-species biofilm, considering variable formation conditions	*Candida albicans* and *Fusarium oxysporum*	The total biomass of the dual-species biofilm was significantly lower in comparison to the single biofilm of *F. oxysporum* but superior to that of the single *C. albicans* biofilm	[142]

Ref—Reference; MFB—Mixed Fungal Biofilm; NDM—non dermatophyte molds; NCAC—Non-*Candida albicans* species; ETT—Endotracheal tube; IGB—intragastric balloons.

## Data Availability

Not applicable.

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
