# Peer review of "Mixed Fungal Biofilms: From Mycobiota to Devices, a New Challenge on Clinical Practice"

_microorganisms, 2022, doi:10.3390/microorganisms10091721_

Round 1
Reviewer 1 Report
de Souza Costa and colleagues have written a review on mixed fungal biofilms. It is very well written and very thoughtful. I have a few comments (minor):
1. P1 line 10: the second line of the abstract need modifying...
2. P2 line 51 positive rather than posterior ....I think
3. P2 L54 .....is based on growth....
4. P2 L70...Two variants...can comprise (rather than compound?)
5. P3 L118 ..has been shown to......I think you need a reference at the end of this important statement
6. P6 L253...Intercession (without s?) point
7. P7 L320 Knowlead ...new word for me!
8. P8 L358?full stop after parapsilosis
9. P8 L363 posteriorly sits uncomfortably in that sentence.
10. P8 L370...in addition (to)
11. P8 L382 ?full stop after IGB
12. P2 L557 (the numbering of pages is a bit odd as it starts from P1 after Table 1!). ?full stop after biofilm
13. P3 L570-571: Permanent cells or persistent cells? need to stick to one
14. P3 L573...?full stop after amp B exposure
15. P3 L590.... of three at 80 times...!
16. Conclusions and perspectives. This needs proof reading.
In addition, I wonder if you need to address the issues of the inadequacy of current microbiological wisdom of labelling 2nd and 3rd organisms as 'contaminants. Do you have a way forward for how this issue should be dealt with?
Author Response
RESPONSES TO REVIEWERS
Title: Mixed fungal biofilms: from mycobiota to devices, a new challenge on clinical practice
Reference number: microorganisms-1869583
Mr. Byron Liu  
Microorganisms
Dear editor and reviewers,
We would like to thank the reviewers for careful and thorough reading of this manuscript and for the thoughtful comments and constructive suggestions, which help to improve the quality of this manuscript. We have studied these comments carefully and have made corresponding corrections that we hope will meet with your approval.
All changes are evidenced with the “Track Changes” function like solicited.
Our response follows (the reviewer’s comments are in italics).
Comments and suggestions from the reviewers to authors:
Reviewer #1:
de Souza Costa and colleagues have written a review on mixed fungal biofilms. It is very well written and very thoughtful. I have a few comments (minor):
- P1 line 10: the second line of the abstract need modifying…
Author's response: Suggestion accepted.
- P2 line 51 positive rather than posterior ....I think
Author's response: Suggestion accepted.
- P2 L54 .....is based on growth....
Author's response: Suggestion accepted.
- P2 L70...Two variants...can comprise (rather than compound?)
Author's response: Suggestion accepted.
- P3 L118 ..has been shown to......I think you need a reference at the end of this important statement
Author's response: Suggestion accepted.
- P6 L253...Intercession (without s?) point
Author's response: Suggestion accepted.
- P7 L320 Knowlead ...new word for me!
Author's response: Suggestion accepted.
- P8 L358?full stop after parapsilosis
Author's response: Suggestion accepted.
- P8 L363 posteriorly sits uncomfortably in that sentence.
Author's response: Suggestion accepted.
- P8 L370...in addition (to)
Author's response: Suggestion accepted.
- P8 L382 ?full stop after IGB
Author's response: Suggestion accepted.
- P2 L557 (the numbering of pages is a bit odd as it starts from P1 after Table 1!). ?full stop after biofilm
Author's response: Suggestion accepted.
- P3 L570-571: Permanent cells or persistent cells? need to stick to one
Author's response: Suggestion accepted.
- P3 L573...?full stop after amp B exposure
Author's response: Suggestion accepted.
- P3 L590.... of three at 80 times...!
Author's response: Suggestion accepted.
- Conclusions and perspectives. This needs proof reading.
In addition, I wonder if you need to address the issues of the inadequacy of current microbiological wisdom of labelling 2nd and 3rd organisms as 'contaminants. Do you have a way forward for how this issue should be dealt with?
Author's response: Authors thank the reviewer for valuable comments, but as far as we know at the moment there are no technical standards (guidelines) published by competent agencies such as (WHO, IDSA, PAHO, ANVISA). In this sense, multicenters and randomized studies will be necessary, carried out with a significant number of patients from different geographic regions, ethnicities, economic conditions and other epidemiological scenarios that vary in humanity, to support a way forward for how this issue should be dealt with. It was suggested in the “Conclusions and Perspectives” on page 18, line 634.
Kind regards,
Dr. Terezinha Svidzinski.

Reviewer 2 Report
In this narrative review the Authors provide a well organized overview of mycotic biofilm
Please discuss rational and limitations of narrative review
I suggest to organize the paper in more paragraphs as this might help the reader to follow
Is biofilm of fungi the same as the biofilm of bacteria?please discuss. A figure representing boil of fungi is suggested.
What about biofilm in mycotic prosthetic infections?
Would the same diagnostic methods used for bacteria biofilm be useful for mycotic ones? Please discuss
Author Response
RESPONSES TO REVIEWERS
Title: Mixed fungal biofilms: from mycobiota to devices, a new challenge on clinical practice
Reference number: microorganisms-1869583
Mr. Byron Liu  
Microorganisms
Dear editor and reviewers,
We would like to thank the reviewers for careful and thorough reading of this manuscript and for the thoughtful comments and constructive suggestions, which help to improve the quality of this manuscript. We have studied these comments carefully and have made corresponding corrections that we hope will meet with your approval.
All changes are evidenced with the “Track Changes” function like solicited.
Our response follows (the reviewer’s comments are in italics).
Comments and suggestions from the reviewers to authors:
Reviewer #2:
In this narrative review the Authors provide a well organized overview of mycotic biofilm.
Please discuss rational and limitations of narrative review.
Author's response: With thanks for your comment, we inform that our narrative overview is a source of educational information, which legibly, joins many reports and published information, evidencing the existence of mixed fungal biofilms formed on biotic and abiotic surfaces. We present a broad perspective on this topic, which unfortunately is still largely neglected.
An important limitation that we could mention would be the scarcity of information on the resources involved in the fungus-fungus interaction in mixed biofilms, which should be of greater interest among researchers considering its complexity and microbiological versatility. Emphasizing that this was due to the scarcity of articles with this focus. This reflection has been introduced in the manuscript on page 18, lines 619-625.
I suggest to organize the paper in more paragraphs as this might help the reader to follow
Author's response: Suggestion accepted.
Is biofilm of fungi the same as the biofilm of bacteria? please discuss. A figure representing boil of fungi is suggested.
Author's response: It is an interesting issue, actually a bacterial or fungal biofilm has structural similarities and both (bacteria and fungi) can even coexist in polymicrobial biofilms. However, they differ in many points, so the discussion for this point would be long and in-depth, and it is not part of the focus of this review, which is centered on the fungal-fungal mixed biofilm, naturally formed on biotic and abiotic surfaces, and their possible repercussion on the clinical management. For the same reason, the stages of formation of a fungal biofilm were not represented illustratively.
What about biofilm in mycotic prosthetic infections?
Author's response: The authors are grateful for the reviewer's suggestion, this device was really missing. We inform you that informations were added:
- In the item “3. What are mixed fungal biofilms and where are they naturally found?” (page 7, lines 343-353).
Periprosthetic joint infections (PJI) usually originate from hematogenous dissemination or surgical site infection. Although fungi represent less than one percent of all reported PJI, the increase in cases described in the last decade draws attention, mainly due to the variability in virulence among fungal species, making treatment unpredictable and challenging. Among the PJI fungal agents, the Candida genus stands out, but is not limited to it [82,83]. A systematic review gathered all the cases already published that bring members of the genus Aspergillus as etiological agents of PJI in 11 reported cases [83]. Highlighting a study including 18 PJI cases that occurred between 2000 and 2015, where 10 of them are mixed infections, nine fungus-bacteria and one fungus-fungus [84]. Despite reports of mixed fungal infections, commonly authors do not raise a discussion about their importance [83,84].
- A reference (171) has been added in table 1.
- In item 5, “Conclusions and Perspectives” on page 19, lines 685-690 have been added with the following comment:
In that line of thought, PJI could be forewarned by a silver-coated titanium treatment, since an in vitro evaluation showed promising results in the prevention of Candida adhesion on titanium [165]. These authors concluded this simple procedure can be a great option in oncological musculoskeletal reconstruction surgeries, highly relevant in patients with highest risk of infectious complications due to immunosuppressive treatments.
Would the same diagnostic methods used for bacteria biofilm be useful for mycotic ones? Please discuss
Author's response: The methodologies used for the diagnosis of microbiological biofilms do not depend on whether it is bacterial or fungal or mixed biofilms involving more than one type of microorganism. The main difference is the culture medium for the growth of fungi or bacteria. However, this issue has not been included in the new version of this manuscript as it is outside the focus of this study.
Kind regards,
Dr. Terezinha Svidzinski.
